

# Integrative approaches to a revision of the liverwort in genus *Aneura* (Aneuraceae, Marchantiophyta) from Thailand

Nopparat Anantaprayoon[1], Passorn Wonnapinij[2] and Ekaphan Kraichak[1,3]

[1] Department of Botany, Kasetsart University, Bangkok, Thailand
[2] Department of Genetics, Kasetsart University, Bangkok, Thailand
[3] Biodiversity Center, Kasetsart University, Bangkok, Thailand

## ABSTRACT

**Background**. The genus *Aneura* Dumort. is a simple thalloid liverwort with cosmopolitan distributions. Species circumscription is problematic in this genus due to a limited number of morphological traits. Two species are currently reported from Thailand, including *A. maxima* and *A. pinguis*. At the global scale, *A. pinguis* is considered a cryptic species, as the species contains several distinct genetic groups without clear morphological differentiation. At the same time, the identity of *A. maxima* remains unclear. In this work, we examined the level of diversity of *Aneura* species found in Thailand using both morphological and molecular data.

**Methods**. We measured the morphological traits and generated the molecular data (four markers: *trnL–F*, *trnH–psbA*, *rbcL,* and ITS2) from the Thai specimens. The concatenated dataset was then used to reconstruct phylogeny. Species delimitation with GMYC, bPTP, ASAP, and ABGD methods was performed to estimate the number of putative species within the genus.

**Results**. The samples of *A. pinguis* formed several clades, while *A. maxima* sequences from Poland were grouped in their clade and nested within another *A. pinguis* clade. We could not recover a sample of *A. maxima* from Thailand, even from the reported locality. Two putative species were detected among Thai *Aneura* samples. However, no morphological trait could distinguish the specimens from the two observed genetic groups.

**Discussion**. The previously observed paraphyletic nature of *A. pinguis* globally was also found among Thai samples, including several putative species. However, we could not confirm the identity of *A. maxima* from Thai specimens. The previous report could result from misidentification and problematic species circumscription within *Aneura*. The results highlighted the need to include multiple lines of evidence for the future taxonomic investigation of the group.

Corresponding author
Ekaphan Kraichak, ekaphan.k@ku.th

# INTRODUCTION

Species delimitation is one of the constant challenges in systematic biology (*Ahmadzadeh et al., 2013*; *Pante, Schoelinck & Puillandre, 2015*). Our limited ability to define clear species boundaries has created a gap in communication across different disciplines, leading to a so-called "taxonomy crisis" (*Dayrat, 2005*). Advances in molecular genetics have allowed the application of the phylogenetic species concept through molecular phylogeny. Systematists can explicitly test the species circumscription with the main criterion of monophyly (*Vanderpoorten & Shaw, 2014*). Many methods have been proposed for species delineation using DNA sequences and molecular phylogeny (*Renner et al., 2017*). For example, the generalized mixed Yule-coalescent (GMYC) model and Poisson Tree Process (PTP) use tree topology and branching patterns to observe significant shifts between conspecific and interspecific processes (*Fujisawa & Barraclough, 2013*; *Pons et al., 2006*; *Zhang et al., 2013*). In addition, sequence-based delimitations like ASAP (*Puillandre, Brouillet & Achaz, 2021*) and ABGD (*Puillandre et al., 2012*) were also proposed to detect the species boundary from genetic distances. These approaches have been applied to various groups of bryophytes, where morphological differences among taxa are limited (*Medina et al., 2012*; *Renner, Brown & Wardle, 2013*).

Despite its utility in identifying cryptic taxa, these delimitation approaches with molecular data have rarely been explored within the family Aneuraceae. The largest family of simple thalloid liverworts, Aneuraceae, comprises 360 taxa from five genera, including *Aneura*, *Riccardia*, *Lobatiriccardia*, *Verdoonia*, and *Afroriccardia* (*Furuki, 1991*; *Preußing et al., 2010*; *Söderström et al., 2016*; *Rabeau et al., 2017*). The delimitation of species in this family has been difficult because only a few morphological characters are available. The key taxonomic characters, like oil bodies, disappeared within a few days after collection, making it impossible to observe from the herbarium specimen (*Bakalin, 2018*).

The genus *Aneura* Dumort. belongs to the family Aneuraceae. Globally, 49 species were reported for the genus (*Söderström et al., 2016*). In Thailand, only two species were reported, including *A. pinguis* (L.) Dumort. from the country checklist (*Lai, Zhu & Chantanaorrapint, 2008*) and *A. maxima* (Schiffn.) Steph. from another report (*Frahm, Pollawatn & Chantanaorrapint, 2009*). Most bryologists recognized that *A. pinguis* is likely a complex of cryptic species, as it has a cosmopolitan distribution and contains multiple polyphyletic groups (*Buczkowska et al., 2016*). The other species, *A. maxima*, was first published from a specimen in West Java, Indonesia (*Schiffner, 1899*) and subsequently reported from other countries in Asia, including India, New Caledonia, Japan, Thailand, and Eastern Russia (*Andriessen et al., 1995*; *Bakalin, 2018*). More recently, this species has also been found from the Appalachian Mountain in North America and many countries in Europe, including Belgium, France, Finland, Denmark, Luxemburg, Czech Republic, Slovakia, Corsica, Germany, Portugal, Spain, Romania, and Norway (*Frahm, 2011*). *Shaw (2001)* proposed the biogeographic patterns among the observed subgroup of *A. pinguis*, similar to *Baczkiewicz, Gonera & Buczkowska (2016)*, who observed the different geographic distribution and habitat between *A. pinguis* complex and *A. maxima*. Moreover, *A. pinguis* and *A. maxima* vary very little in their morphological characters. The specimens of

*A. maxima* are generally large than the *A. pinguis* with undulate and translucent margins composed of 5–15 cells of an unistratose wing, while *A. pinguis* sample had crispate and opaque margins formed by 1–4 cells of an unistratose wing (*Bakalin, 2018*). None of the studies, however, have included a specimen of *A. maxima* from Indonesia, the locality of the first publication, making it difficult to verify its taxonomic status. Because of the completely overlapped distribution and morphological similarity, these two species of *Aneura* might belong to the same species complex.

Additional evidence for the *Aneura pinguis* complex came from chemical and molecular studies on this group. *Szweykowski & Odrzykoski (1990)* first showed several groups of *A. pinguis* using enzymatic markers and suggested that habitat differences might be responsible for the observed groups. *Buczkowska, Adamczak & Baczkiewicz (2006)* reported minor differences in thallus thickness and the number of cells in the middle of thallus among the examined specimens, which were later corroborated by the differences in the volatile compound composition between *A. pinguis* and *A. maxima* (*Wawrzyniak et al., 2014*). After that, *Buczkowska et al. (2016)* reported more molecular evidence using the ISSR method showing several cryptic species within *A. pinguis*, and eventually found chemotaxonomic markers congruent with the genetic groups with *A. pinguis* from DNA barcoding (*Wawrzyniak et al., 2021*). Finally, molecular phylogeny using five chloroplast regions (*matK, rbcL, rpoC1, trnH–psbA,* and *trnL–trnF*) and the nuclear ITS region (ITS1-−5.8S–ITS2) showed that *A. pinguis* contained ten paraphyletic taxa with *A. maxima* and *A. mirabilis* nested among these clades (*Baczkiewicz et al., 2017*). These studies demonstrated problematic species delimitation in the genus *Aneura*. However, most of the studied specimens came from temperate Europe with little to no representative from the other regions where *Aneura* can be found, including Thailand.

This manuscript presents the results of integrative approaches to a revision of the genus *Aneura* in Thailand in the context of genetic diversity. Our specific objectives were: (1) to determine the phylogenetic position of *Aneura* specimens from Thailand relative to known global samples, (2) to identify genetically distinct clades within the genus using both tree-based and sequence-based delimitation methods, and (3) to evaluate the efficacy of available morphological data for delimiting observed genetic clades within the genus. By integrating multiple lines of taxonomic evidence and delimitation methods, we aimed to provide a more precise estimate of the *Aneura* diversity in Thailand and to establish a foundation for future investigations of this genus in the region.

## MATERIALS & METHODS

### Plant materials and identification

According to previous reports, fresh specimens of *Aneura* were collected in various protected areas, including Doi Inthanon National Park, Khao Yai National Park, and Phu Luang Wildlife Sanctuary. The collection was permitted by the Department of National Parks, Wildlife, and Plants MNRE 0907.4/1075. Dry herbarium specimens were obtained from bryophyte herbaria in Thailand, including BCU (Chulalongkorn University), CMUB (Department of Biology, Chiang Mai University), PSU(Prince of Songkla University),
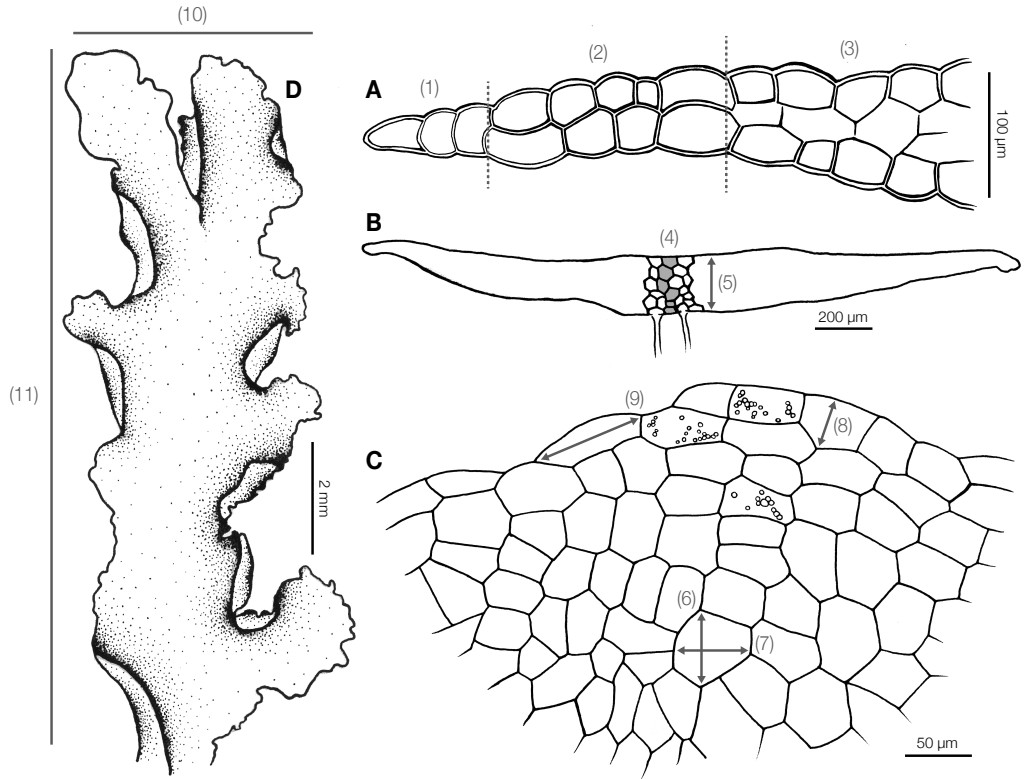

**Figure 1** **Diagram illustrating morphological and anatomical measurement of *Aneura* specimens.** (A) (1) Unistratose cells (2) bistratose cells (3) tristratose cells; (B) (4) number of maximum cell layers (5) thallus thickness; (C) (6) median cell width (7) median cell length (8) marginal cell width (9) marginal cell length; (D) (10) thallus width (11) thallus length; (1) to (5) were measured from the cross section, and (6)–(11) were measured from dorsal view.

and Herbarium of the Department of Biology, Burapha University. The specimens were identified to the genus to confirm its affinity to *Aneura*, using relevant taxonomic literature at the family rank for avoiding a bias of species identification. The specimens were dried and preserved in paper packets as voucher specimens deposited in the herbarium at Kasetsart University.

## Morphological and anatomical measurement

Before morphological and anatomical measurement, dry specimens were rehydrated on moist paper and hand cut under a stereomicroscope. The whole thallus and cross sections were examined under a stereomicroscope and a light compound microscope. We measured 13 quantitative (Fig. 1) and 20 qualitative characters (Table S1) for subsequent statistical analyses. Continuous characters were measured from photographs in the Fiji software (*Schindelin et al., 2012*) with a stage micrometer for scale calibration. Qualitative characters were recorded as presence or absence.

## DNA extraction, amplification, and sequencing

Genomic DNAs were extracted from fresh thallus, dried plant materials, and herbarium specimens using Nucleospin® Plant II Kit(Macherey-Nagel Gmbh & Co). The extraction followed the manufacturer's protocol with exceptions for adding 10 µL of RNase A, PL2 lysis, and using a single wash before elutions.

In this study, we amplified four DNA markers, using the following primers: (1) for *trnL-trnF*, markers: trnL–trnF–C (5′–CGAAATCGGTAGACGCTACG–3′) and trnL"—trnF–F (5′–ATTTGAACTGGTGACACGAG–3′) (*Taberlet et al., 1991*), (2) for *trnH-psbA*, psbA501F (5′–TTTCTCAGACGGTATGCC–3′) and trnHR (5′–GAACGACGGGAATTGAAC–3′) (*Forrest & Crandall-Stotler, 2004*), (3) *rbcL*, rbcL–a–F (5′–ATGTCACCACAAACAGAGACTAAAGC–3′) and rbcL–1200–R(5′–TGYCCYAAAGTTCCACCACC–3′) (*Gradstein et al., 2006*; *Kress & Erickson, 2007*), and (4) ITS2, Bryo5.8SF (5′–GACTCTCAGCAACGGATA–3′) and Bryo26SR (5′–AGATTTTCAAGCTGGGCT–3′) for ITS2 (*Hartmann et al., 2006*).

Each PCR reaction contained 25.5 µL, including 9.5 µL of Ultra OnePCR™ reaction mixture (Bio-Helix Co. Ltd), 9.5 µL of nuclease-free water, 2.5 µL of each primer at 10 µM and 1.5 µL of DNA extractions. The PCR conditions were as follows: (1) for *trnL–F*, initial denaturation for 2 min at 92 °C, followed by 30 cycles of 1 min at 92 °C, 50 s at 51 °C, 90 s at 72 °C, and final elongation for 10 min at 72 °C, (2) for *trnH–psbA*, initial denaturation for 4 min at 94 °C, followed by 30 cycles of 1 min at 94 °C, 30 s at 62 °C, 1 min at 72 °C, and final elongation for 7 min at 72 °C, (3) for *rbcL*, initial denaturation for 4 min at 94 °C, followed by 30 cycles of 1 min at 94 °C, 50 s at 51 °C, 90 s at 72 °C, and final elongation for 10 min at 72 °C, and (4) for ITS2, initial denaturation for 75 s at 94 °C, followed by 35 cycles of 35 s at 95 °C, 55 s at 55 °C, 42 s at 72 °C, and final elongation for 10 min at 72 °C. PCR products were loaded on 1% agarose gel using GenedireX Agarose Tablets and visualized under an LED transilluminator to observe the products. Successful products were purified with ExoSAP-IT PCR Product Cleanup(Applied Biosystems, Santa Clara, California, USA) for 15 min at 37 °C and 80 °C, respectively. Cleaned products were sent for bidirectional sequencing at Macrogen Inc. in South Korea (http://www.macrogen.com/).

## Sequence assembly and phylogenetic reconstructions

Geneious Prime 2022.2.1 (https://www.geneious.com) was used to assemble nucleotide sequences for each region, which were then aligned with available sequences of Aneuraceae from the NCBI nucleotide database (Table S2). Sequences were aligned using the Geneious Alignment tool, and the cost matrix was set to 70% similarity. The resulting alignments were adjusted manually and then concatenated using MEGA X (*Kumar et al., 2018*). The resulting alignments were uploaded to CIPRES (http://www.phylo.org) (*Miller, Pfeiffer & Schwartz, 2010*) and analyzed using Maximum Likelihood (ML) and Bayesian Inference (BI). The ML analysis was run using the RAxML–HPC BlackBox version 8.2.12 (*Stamatakis, 2014*) program with the bootstrap sampling for 10,000 pseudo-replications and the GTR+I+G as a substitution model. Single-locus and concatenated datasets were analyzed, and a node with a support value greater than or equal to 70 percent was regarded as well-supported. The BI analysis was conducted using MrBayes on XSEDE (3.2.7a) (*Ronquist et*

*al., 2012*) with the GTR+I+G substitution model. Sampling trees were estimated using the Markov chain Monte Carlo (MCMC) method, with four chains running for 20,000,000 generations and sampling every 1,000th tree. The first 4,000 trees were discarded with the majority rule consensus tree as burn-in. The effective sample sizes(ESS) were all greater than 200, as determined using Tracer v1.7.1 (*Rambaut et al., 2018*). A posterior probability greater than or equal to 0.95 was considered strong support for a node in a BI tree. Phylogenetic outputs were visualized using FigTree version 1.4.4 (*Rambaut, 2012*) to identify any conflicts in topologies among the reconstructed trees. Finally, reconstructed phylogenetic trees were created using R environments (*R Core Team, 2019*; *RStudio Team, 2020*) with the 'ape' package (*Paradis, Claude & Strimmer, 2004*) and visualized with the 'ggtree' package (*Yu, 2020*).

## Species delimitation analysis

To identify putative species units within the genus *Aneura*, we applied tree-based and sequence-based species delimitation to our phylogenetic trees and concatenated dataset. For tree-based species delimitation, the ML tree of the concatenated dataset was used with outgroups from the members of *Riccardia* and *Lobatiricardia* using the drop.tip command in 'ape.' The tree was then transformed into an ultrametric tree, using Penalised Likelihood with the 'chronos' command. The first tree-based method, the generalized mixed Yule-coalescent (GMYC) model, was performed to examine putative species boundaries and detect the position of the shift between intraspecific and interspecific diversification, using the function 'gmyc' from the R package 'splits' (*Ezard, Fujisawa & Barraclough, 2009*). The second tree-based method was the Bayesian Poisson Tree Process(bPTP). The same ultrametric tree from the first method was subjected to a run of 100,000 MCMC generations and thinning value of 100 through their web server (https://species.h-its.org/ptp/). For the sequence-based method, the concatenated dataset was subjected to the Assemble Species by Automatic Partitioning (ASAP) model (https://bioinfo.mnhn.fr/abi/public/asap/asapweb.html) and the Automatic Barcode Gap Discovery (ABGD) model (https://bioinfo.mnhn.fr/abi/public/abgd/abgdweb.html) on their respective web servers. The Kimura 2-parameter (K80 or K2P) model and other default parameters were used as the substitution model.

Due to the limited access to the specimens, morphological and anatomical characters were compared between the observed genetic groups only for specimens from Thailand. For each quantitative character, we performed non-parametric Wilcoxon Rank Sum and Signed Rank Tests (*Hollander, Wolfe & Chicken, 2013*) with the 'wilcox.test' command in the R package 'stats.' For qualitative character, Pearson's chi-squared test (*Patefield, 1981*) was applied to the tabulated contingency table with the 'chisq.test' command in the R package 'stats.' A $p$-value less than or equal to 0.05 was considered a significant difference between the compared groups. These characters were also subjected to the principal components analysis (PCA) method with the 'prcomp' command in R. Prior to PCA, the qualitative data set was transformed to a distance matrix as the Jaccard index, using the 'vegdist' command in the R package 'vegan' (*Oksanen et al., 2019*) to reduce the false similarity due to joint absence. Multidimensional space plots from PCA were displayed

using the 'autoplot' command in the R package 'ggfortify' (*Tang, Horikoshi & Li, 2016*). The PERMANOVA was perform to determine the multivariate differences among the two observed genetic groups, using the 'adonis2' command in the R package 'vegan'.

## RESULTS

We generated 115 new sequences and aligned them with available sequences of Aneuraceae in the NCBI database (Table S2). Most of the NCBI sequences came from the specimens in Poland in previous publications by *Preußing et al. (2010)* and *Baczkiewicz et al. (2017)*. The final multilocus dataset contained 2,006 nucleotide positions. From the B/MCMC analysis, we obtained the mean log of the likelihood at $-15,001.7575$, while the final ML optimized tree had the likelihood of $-14,999.720835$. The topologies of the ML and BI analysis showed no significant conflicts. Therefore, we displayed only the topology from the ML analysis along with the bootstrap support value and posterior probability (Fig. 2).

The concatenated phylogenetic tree revealed several clades within the *Aneura* samples (Fig. 2). The *A. pinguis* specimens from Thailand did not form a monophyletic clade but were placed throughout the tree among the specimens from Japan, Ireland, and Poland in the GYMC groups 1 and 4 (Fig. 2). The sequences of *A. maxima* from Poland formed their monophyletic clade nested within other *A. pinguis* clades. The specimens SCOS2695 and SCOS1570 were identified initially as *A. maxima* but were placed among the other sequences of *A. pinguis*. The larger specimens of *Aneura*, including the specimen that was originally identified as *A. maxima* from Indonesia, formed a clade with the samples of *Lobatiriccardia* species, separated from the rest of the genus *Aneura*.

When applied to the tree of the genus *Aneura*, the GMYC method found a single threshold that divided the genus into four putative species (Fig. S1). The likelihood of the GYMC model was not significantly different from the null model ($L_{GMYC} = 192.4754$, $L_{null} = 191.5813$, $P_{LRT} = 0.4089$). The four putative species comprised three clades of *A. pinguis* specimens and the clade of *A. maxima* from Poland. The first putative species included most of the Thai specimens in this study, while the second putative species contained exclusively the sequences from Poland. The final putative species included sequences from Poland, Japan, Ireland, and Thailand. Most of these clades identified by GMYC were not strongly supported.

The other delimitation methods resulted in a larger number of putative species than GMYC. The bPTP method yielded 45 putative species, many of which were singletons (acceptance rate = 0.7295, merge = 50209, split = 49791). For ASAP, the best partition had the ASAP score of 2.00 at the threshold distance at 0.223 and $P_{ASAP} = 0.136$. The ABGD method used the initial partition with prior maximal distance at $P = 0.00774$ and the barcode gap distance at 0.020 with $P_{ABGD} = 0.008$ for its delimitation. The ASAP and ABGD methods recovered the same 28 putative species, one of which was the *A. maxima* clade from Poland, similar to the GMYC method. Because the smaller putative species identified in bPTP, ASAP, and ABGD could be a group within a larger clade identified in GMYC, we used the putative species from GMYC for morphological comparisons.

The specimens from Thailand were placed in the putative species or clades number 1 and 4. The specimens in these two specimens showed no significant differences in their

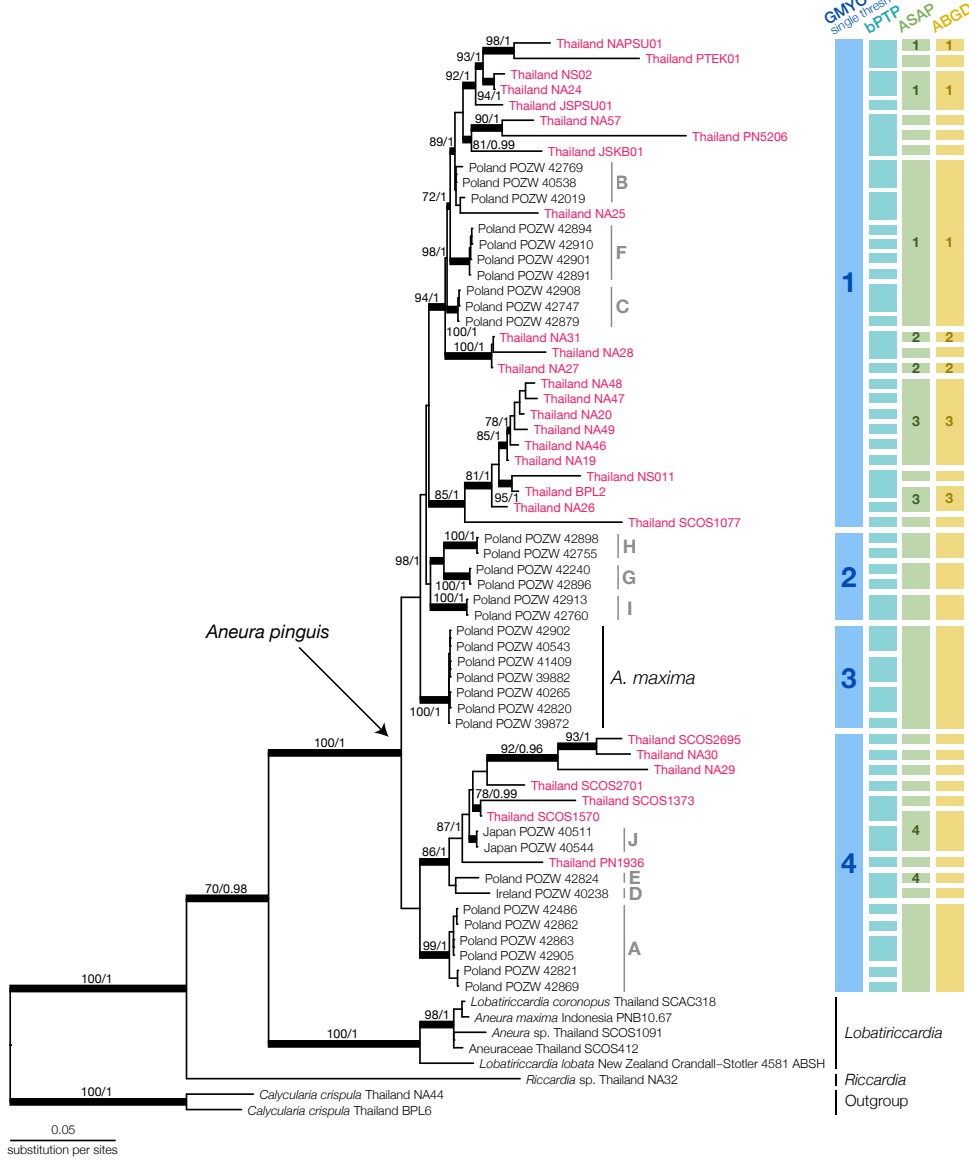

**Figure 2 Maximum Likelihood tree of *Aneura* species from the concatenated dataset of ITS2, *trnL-trnF*, *trnH-psbA* and *rbcL* and the results of species delimitation using GMYC, bPTP, ASAP, and ABGD methods.** Red characters indicate Thai *Aneura* samples we found in this study, with light-grey groupings indicate Poland's *Aneura* situation from previous studies. All thickened branches indicate the well-support nodes in either Maximum Likelihood (ML) or Bayesian Inference (BI). The numbers on the branches represent the bootstrap support values from ML and posterior probability from BI, respectively.

continuous traits ($P_{\text{wilcoxon}} \geq 0.051$, Fig. S2) and discrete traits ($P_{chi} \geq 0.33$, Fig. S2). Likewise, the principal component analysis and the results of PERMANOVA showed completely overlapping morpho-spaces of these two putative species in both qualitative and quantitative traits ($P \geq 0.37$, Fig. 3).

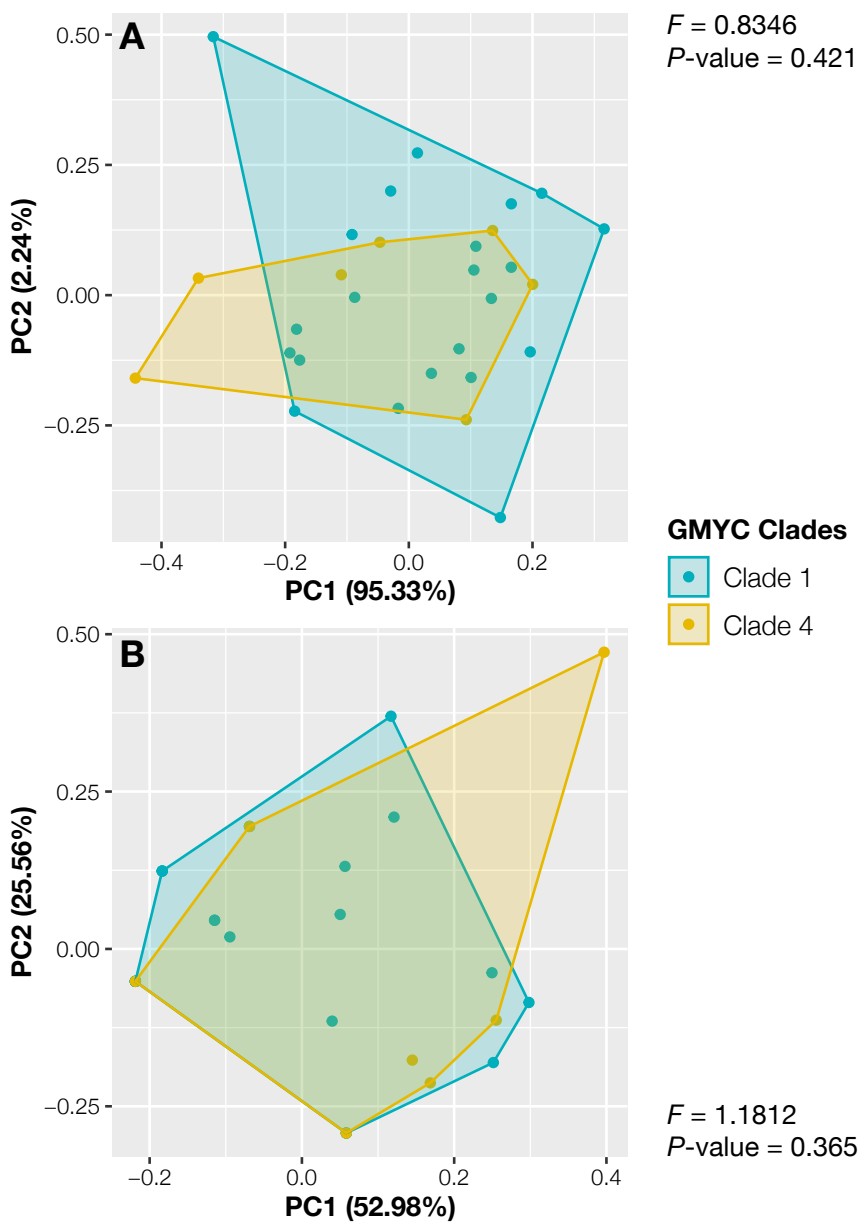

**Figure 3** **Principal component analysis of morphological traits of Thai *Aneura* specimens.** (A) Continuous traits and (B) discrete traits. The colors of data points and polygons represent two detected clades from GMYC. The F-values and *P*-values from PERMANOVA results showed no significant differences between two GMYC clades ($P \geq 0.0365$, Clades 1 and 4).

## DISCUSSION

### Backbone relationships of Thai *Aneura* species

The *Aneura* specimens from Thailand revealed a comparable level of phylogenetic diversity to those at the global scale. The specimens of *Aneura pinguis* in Thailand were placed in

several clades interspersed with the samples from the other regions, suggesting dispersals of multiple genetic groups into the country instead of *in situ* speciation. Multiple putative species within the *A. pinguis* were consistently recovered from several molecular markers, including polymerase chain reaction-restriction fragment length polymorphism (PCR-RFLP) test (*Wachowiak et al., 2007*), ISSR markers determination (*Buczkowska et al., 2016*), and genomic DNA (*Baczkiewicz et al., 2017*). The analysis also confirmed the paraphyletic nature of *A. pinguis*, observed in the previous study (*Baczkiewicz et al., 2017*). A recent study sequenced several specimens from the locality nearby the original description of *Jungermannia pinguis* L., the basionym of *A. pinguis* (L.) Dumort., to provide the lectotype and the epitype for the name (*Long et al., 2023*). The BLAST searches of the sequences from the study showed the sequences of the epitype (BOLDSYSTEMS Accession No. LWT535-21) were 100% identical to those from the specimen POZW:42901 (the Clade F, GMYC) in our study, suggesting that Thai specimens of *A. pinguis* belong to the same genetic group as the originally described *J. pinguis* from England.

The *A. maxima* specimens from Poland were nested among other *A. pinguis* clades. Another study also found *A. maxima* and *A. mirabilis* ( =*Cryptothallus mirabilis*) to be nested within *A. pinguis* clades, despite the apparent morphological and physiological differences (*Wickett & Goffinet, 2008*). The evidence supports the idea that *A. pinguis* contains multiple putative species, many of which are also found in Thailand.

## Cryptic species within *Aneura pinguis*

The results provided evidence for multiple cryptic species within *Aneura pinguis*. All chosen species delimitation methods detected more than one putative species among studied *Aneura* sequences, but morphological characters were not significantly different among the putative species. Most of the chosen methods identified a large number of putative species, between 28 to 45 species, indicating over-estimation or over-splitting of the species units, while the GMYC method only recovered four putative species, or at least detected the transition from speciation and coalescent processes (*Papadopoulou et al., 2009*). We chose to compare the morphological characters among the putative species from the GMYC method, as it appeared to perform relatively well in various groups of organisms, such as gastropods, frogs, bats, and chimpanzees (*Pons et al., 2006*; *Carstens et al., 2013*; *Satler, Carstens & Hedin, 2013*; *Malavasi et al., 2016*). Likewise, some liverwort cryptic species had been explored using organellar genome in the species delimitation. For example, the super-barcoding, especially with the complete mitochondrial genome, potentially has the ability to identify the species within the cryptic lineages of Pellidae and *Calypogeia* (*Ślipiko et al., 2020*; *Ślipiko et al., 2022*; *Paukszto et al., 2023*). The observed different outcomes of species delimitation methods might be due to small genetic differences among the putative species, as evident from the negative barcoding gap in the previous study (*Wawrzyniak et al., 2021*). Regardless of the delimitations, it was clear that at least two putative species of *A. pinguis* occurred in Thailand.

We could not distinguish the two putative species of *A. pinguis* from the morphological characters in Thai specimens. These two putative species could result from the cryptic speciation within the *A. pinguis* complex. A recent molecular study showed that

mitochondrial and plastid genomes of *A. pinguis* demonstrated a rapid evolutionary rate than other liverworts (*Myszczyński et al., 2017*). *Shaw (2001)* also suspected that several cryptic species within the *A. pinguis* complex could come from many isolated "microallopatric" populations of this cryptic species throughout Europe and North America. However, no detailed population genetics study has been conducted to verify this kind of speciation within the species.

The other possibility for overlapping morphological characters could be phenotypic plasticity in bryophytes. The thallus morphology of *Aneura* is more variable than sporophyte characters, such as seta, elater, capsule structure, and spore (*Baczkiewicz et al., 2017*). While it could be more desirable to use sporophytic characters for classification, sporophytes are relatively rare and short-lived among the members of Aneuraceae (*Preußing et al., 2010*; *Reeb et al., 2018*). The morphology of individual thallus can be the product of the interaction between genetics and ecological differences (*Renner, Brown & Wardle, 2013*). The largest difference between the observed putative species was the thallus width, one of the main characters separating *A. maxima* from *A. pinguis*. However, each clade contained specimens from various localities that potentially had different environmental conditions. Therefore, the genetic differences among the clade may not be enough to overcome the phenotypic plasticity of the gametophyte. A common garden experiment of different genotypes will be required to test this hypothesis in the future. Any further nomenclatural actions to these putative species will require more precise morphological differentiation among the clades to ensure that the named taxa are distinguishable for taxonomists without molecular evidence.

## Status of *A. maxima* in Thailand

We could not confirm the occurrence of *Aneura maxima* in Thailand in our survey of fresh specimens and herbarium vouchers. The first occurrence of *A. maxima* in Thailand was reported from Khao Yai National Park (*Frahm, 2011*), where we made several trips to the same locality and could not recover a specimen of *A. maxima* in any part of this national park. Two herbarium specimens, SCOS2695 and SCOS1570, which were identified by the collectors as *A. maxima* due to their noticeably larger thallus than typical *A. pinguis*. However, the molecular phylogenetic analysis placed these two accessions among other samples of *A. pinguis*. Because of that, we designated the name of *A. pinguis* to these two specimens (SCOS2695 and SCOS1570). Additionally, the specimen PNB10.67 from West Java (Indonesia) was identified as *A. maxima* but placed with the specimens of *Lobatiriccardia* in the molecular phylogeny. Misidentification of these specimens as *A. maxima* could result from unusually large thallus size. Thus, upon our close examination, these specimens did not match the description of the reported *A. maxima* (*Andriessen et al., 1995*; *Bakalin, 2018*; *Buczkowska & Baczkiewicz, 2006*; *Frahm, 2011*; *Frahm, 1997*; *Loskotová, 2006*). Given the evidence, we proposed that the previous reports of *A. maxima* in Thailand were most likely the misidentification of a variation of *A. pinguis* in the country and all the specimens we found in this study should be the *A. pinguis*. The specimen of the supposed "*A. maxima*" deposited at BONN has to be located and examined to verify the status of *A. maxima* in the country.

The confusion over *A. maxima* and *A. pinguis* was not surprising. Originally, the genus *Aneura* adopted a more inclusive circumscription and included three subgenera, including *Aneura* Dumort., *Lobatiriccardia* Mizut. & Hatt., and *Austroaneura* Schust. (*Wachowiak et al., 2007*). Since 1991, the *Aneura sensu lato* have become two individual genera, including *Aneura* Dumort. and *Lobatiriccardia* (Mizut. & Hatt.) Furuki, based on branching patterns and oil-bodies forms (*Furuki, 1991*). Oil bodies are often difficult to observe in the herbarium specimens, as they tend to disappear within a few days after collecting. Therefore, our ability to distinguish between a large-thallus *Aneura* and *Lobatiriccardia* can be limited. Moreover, despite being first described from Indonesia (*Schiffner, 1899*), a known *A. maxima* specimen from the type locality has not been sequenced to determine the status of the species. The *A. maxima* specimens reported from Europe and North America are likely different taxonomic entities from those from Asia.

## CONCLUSIONS

By integrating molecular phylogenetics with morphological data, our study clarifies the diversity level of the liverwort genus *Aneura* in Thailand. At least two putative species within the *A. pinguis* were discovered in Thailand, reflecting the multiple known lineages of this group at the global scale. The occurrence of *A. maxima* in Thailand could not be verified, as all *A. maxima* specimens were either placed with *A. pinguis* or the members of *Lobatiriccardia*. This result also provided additional evidence for the ongoing issue with the status of *A. maxima*, where no DNA sequence was produced from the samples in the type locality. Further studies should look further into population genetics and possible ecological and geographical factors at the global scale, to untangle the taxonomic issue of this genus of simple thalloid liverwort.

## ACKNOWLEDGEMENTS

We would like to thank the following colleagues for their support with the specimens: Dr. Sahut Chantanaorrapint and Dr. Jiroat Sangrattanaprasert from the Department of Biology, Prince of Songkla University, Dr. Phiangphak Sukkharak from the Department of Biology, Burapha University, and Dr. Narin Printarakul from Department of Biology, Chiang Mai University for their help with the specimens. We also thank Dr. Prasart Kermanee from the Department of Botany, Kasetsart University, for his expertise and guidance in botanical illustrations. Lastly, we would like to thank Dr. Steven D. Leavitt from the College of Life Sciences, Brigham Young University, for his expertise and guidance in the species delimitation analysis.

### Funding

Nopparat Anantaprayoon's graduate program was supported by the Graduate Program Scholarship from the Graduate School, Kasetsart University. This project was financially supported by the Office of the Ministry of Higher Education, Science, Research, and

Innovation and the Thailand Science Research and Innovation through the 2021 Kasetsart University Reinventing University Program, Thailand, and by the National Research Council of Thailand (NRCT), 2022 annual funding under Thailand Science Research and Innovation (TSRI). The funders had no role in study design, data collection and analysis, decision to publish, or preparation of the manuscript.

### Grant Disclosures

The following grant information was disclosed by the authors:
Graduate Program Scholarship from the Graduate School, Kasetsart University.
Office of the Ministry of Higher Education, Science, Research, and Innovation.
Thailand Science Research and Innovation through the 2021 Kasetsart University Reinventing University Program, Thailand.
National Research Council of Thailand (NRCT), 2022 annual funding under Thailand Science Research and Innovation (TSRI).

### Competing Interests

The authors declare there are no competing interests.

### Author Contributions

- Nopparat Anantaprayoon conceived and designed the experiments, performed the experiments, analyzed the data, prepared figures and/or tables, authored or reviewed drafts of the article, and approved the final draft.
- Passorn Wonnapinij conceived and designed the experiments, authored or reviewed drafts of the article, and approved the final draft.
- Ekaphan Kraichak conceived and designed the experiments, performed the experiments, analyzed the data, authored or reviewed drafts of the article, and approved the final draft.

### Field Study Permissions

The following information was supplied relating to field study approvals (*i.e.*, approving body and any reference numbers):

The collection was permitted by the Department of National Parks, Wildlife, and Plants MNRE 0907.4/1075

### DNA Deposition

The following information was supplied regarding the deposition of DNA sequences:

The ITS sequences are available in GenBank: OQ708354–OQ708374.

### Data Availability

The raw data are available in the Supplemental Files.

### Supplemental Information

Supplemental information for this article can be found online at http://dx.doi.org/10.7717/peerj.16284#supplemental-information.

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
