# Peer review of "Integrative approaches to a revision of the liverwort in genus Aneura (Aneuraceae, Marchantiophyta) from Thailand"

_PeerJ, doi:10.7717/peerj.16284_

## Round 0.1 · original submission · Major Revisions

Fortunately, and after waiting a few weeks all reviewers finally submitted their comments. Therefore, you have a lot to consider, to take into account the suggestions of the reviewers. The main concerns are the lack of clarity on how to recognize species from a phylogeny, therefore your methods are not suitable to recognize the status of Aneura maxima. On the other hand, you have to review the correct identification of some species. Finally there are some interesting perspectives such like ecological factors to explain the phylogeny. Two of the reviewers made directy their comments in the attached files. In your rebuttal letter you have to include all of them, from the files and from the comments below.

Reviewer 1 ·

Basic reporting

My comments are in the attached pdf.

Experimental design

Methods were described with sufficient details—further comments are in the attached pdf.

Validity of the findings

My comments are in the attached pdf.

Additional comments

My comments are in the attached pdf.

Annotated reviews are not available for download in order to protect the identity of reviewers who chose to remain anonymous.

·

Basic reporting

The article is written in an unambiguous, professional manner. The introduction and background are sufficiently provided with literature references.

Experimental design

Research objectives are well defined. Methods sufficiently described in detail.

Validity of the findings

The conclusions are well stated.

·

Basic reporting

The paper by Anantaprayoon et al. provides novel data on distribution of species of Aneura pinguis complex in Thailand. Since most of the previously studied species were of American or European origin, the presented results are important for better understanding evolutionary processes within the Aneura genus. Moreover, the Authors have not confirmed the presence of A. maxima in Thailand.I’m not surprised by the failure of morphological based approach to deliminate studied Aneura species - these species were studied since 90’s, including in vitro cultivation and common garden experiments, and so far, any morphological markers wasn’t developed. The manuscript is generally well written however I found some points needing attention and further improvements.

Experimental design

Introduction
line 96, please italicize species names.
Lines 101-102, “global diversity” is too strong in this case, however it would be great if authors could include more records - some of them have to be extracted from complete plastome sequences.

Methods:
I would not recommend merging ITS2 and cpDNa dataset, since the incongruence between those two regions was previously raised (Bączkiewicz et al. 2017). It could lower the BI and bootstrap scores and impact molecular delimitation tests.
Using the cryptic species lineage letters during result description and discussion (as well as on Figure 2) would improve readability, especially in the context of previous studies.

Validity of the findings

Results

Line 254: Please use letters instead on number to describing Aneura lineages - it will improve comparison with previous studies (Long et al. 2023 https://doi.org/10.24823/EJB.2023.1932, Bączkiewicz et al. 2017, Myszczyński et al. 2017).
The inclusion output trees of GMYC and ASAP analyses would improve the result section and will make it less descriptive.

The results will also gain more impact if obtained sequences will be analyzed in the wider context and include complete dataset from Baczkiewicz et al. 2017 and Long et al. 2023). It in my opinion, would better visualize the evolutionary position of samples from Thailand.

Discussion

“Cryptic species within Aneura pinguis” paragraph could be improved by broadening discussion about liverworts, not animals (lines 283-284). The GMYC, ASBP and bPTP methods were applied to delimitation species and cryptic species within Calypogeia and Pellidae taxa (Ślipiko et al. 2020 https://doi.org/10.1186/s12870-020-02435-y, Ślipiko et al. 2022 https://doi.org/10.3390/ijms232415570, Paukszto et al. 2023 https://doi.org/10.1038/s41598-023-35269-3). I can hardly agree with statements in lines 285-286.The genetic differences among Aneura species are rather high in comparison to other liverworts species at generic level as author stated in lines 292-294.

Reviewer 4 ·

Basic reporting

.

Experimental design

.

Validity of the findings

.

Additional comments

The review is attached.

Annotated reviews are not available for download in order to protect the identity of reviewers who chose to remain anonymous.

Reviewer 5 ·

Basic reporting

should be improved, see below

Experimental design

the data on morphology, ecology and distribution of the taxa should be added, mostly see below

Validity of the findings

conclusions are the simple repetition of already known things; also see below

Additional comments

The work is devoted to the study of the genus Aneura in Thailand. Unfortunately, the results obtained are rather small. In fact, these are several new sequences built into the dataset already available in the Genbank. The authors need to think about how to make the study more relevant to the reader and avoid repeating long-known facts about the complex nature of Aneura pinguis.

Below I placed some small remarks and general summary

Line 28 “The samples of A. pinguis formed several paraphyletic clades” probably means ‘several clades’, because the clade cannot be paraphyletic.
Line 57-58 “The largest family of simple thalloid liverworts, Aneuraceae, comprises 360 taxa from five genera (Furuki, 1991; Preußing et al., 2010).” – I guess authors should refer to World liverwort checklist (Söderström et al., 2016), not to literature sources where the counting of the taxa was not the primary task.
Line 65 “In Thailand, only two species were reported, including A.pinguis (L.) Dumort. from the country checklist (Lai et al., 2008)”. The checklist is not the primary source for the report of A. pinguis, please cite original. Cite several papers, if necessary.
Line 70 “and subsequently reported from other countries in Asia, including India, New Caledonia, Japan, Thailand, and Eastern Russia)” – the reference are necessary for these reports
Line 233-235 “The larger specimens of Aneura, including the sample of A. maxima from Indonesia, formed a clade with the samples of Lobatiriccardia species, separated from the rest of the genus Aneura.” -- this means (together with Fig. 2 of the ms.) that the only specimen named as A. maxima from Indonesia is simple misidentification of Lobatiriccardia coronopus (or the contamination). Therefore no Aneura specimens from Indonesia were involved into the study.
Line 316-319: “Two herbarium specimens, SCOS2695 and SCOS1570, were identified by the collectors as A. maxima due to their noticeably larger thallus than typical A. pinguis. However, the molecular phylogenetic analysis placed these two accessions among other samples of A. pinguis.” -- The thallus width is not the differentiation feature between these taxa. The differentiation features are in the shape of the thallus margin (undulate) and the width of the unistratose thallus wing. Did you restudy the specimen morphology or you believed to collectors completely?
Line 324-326 “Given the evidence, we propose that the previous reports of A. maxima in Thailand were most likely the misidentification of a large variation of A. pinguis in the country.” – This suggestion is completely unfounded due to: a) authors do not know what is true A. maxima, because they had only one misidentified Lobatiriccardia from Indonesia instead true A. maxima and b) nobody know whether specimens named as A. maxima from Poland are actually identical with true A. maxima described from indonesia. Moreover, authors wrote definitely at the lines 337-339 “The A.maxima specimens reported from Europe and North America are likely different taxonomic entities from those from Asia.”

Authors overlooked the paper on Aneura (Long et al., 2023), where the type species, A. pinguis, was typified and the specimen collected from near the type locality was sequenced. The comparison the sequences obtained by the authors of the present manuscript with sequences from Long et al., 2023 would show which clade of the several ‘A. pinguis’ clades is actually correspond to the true A. pinguis; and if ‘true’ A. pinguis is distributed in Thailand. Please look at https://journals.rbge.org.uk/ejb/article/view/1932

Summing up:
The purpose and result of the work carried out remained unclear to me. The authors showed that A. pinguis is a complex cryptic species, but this was known before them and there is no novelty here. All arguments regarding the occurrence (or the absence) of A. maxima in Thailand do not make sense, because the authors of the paper did not have specimens of A. maxima from Indonesia (at least from Java Island from where it was described) for genetic studies and they did not compare the type specimen of A. maxima (from Java) with Thailand vouchers morphologically . Authors did not provide any morphological or genetic real evidences in favor or against the point of view if the true A. maxima occurs in Thailand. The European materials can hardly be identified with Indonesian ones (and the authors write about this very correctly, but this is not news). What then is the result of the article? Show that Aneura material from Thailand is genetically heterogeneous? But this could be assumed. It is regrettable that the authors stopped at this statement, and did not go further, to an attempt to distinguish "cryptic species" or their complexes using morphological or ecological characters or distribution patterns.

I believe that the article should be rewritten, concentrating on the analysis of ecological, and, perhaps, altitudinal patterns of distribution of representatives of each clade. This will be the first step towards explaining the existing genetic diversity in terms of evolutionary geography and ecology. In addition, it would be appropriate to give an outline of the geographical distribution of each "cryptic species" on the basis of available materials, with detailed maps for the territory of Thailand. Otherwise, the article makes no sense.

---

## Round 0.2 · Minor Revisions

The four reviewers for this round of review coincided with a few changes to improve the manuscript. Please consider them; one included suggestions in the attached file.

Reviewer 1 ·

Basic reporting

Sufficient

Experimental design

Well-defined

Validity of the findings

All necessary data have been provided

Reviewer 4 ·

Basic reporting

The authors have corrected their manuscript very well, taking into account the comments of the reviewers. Apart from a minor comment on some missing reference references, I have no major comments on this version.

It is a really good paper, helpful to the community, highlighting the difficulty of species delimitation. Perhaps you could have discussed the interest in naming, or even delineating cryptic species that can only be distinguished with a complete mitochondrial genome or chloroplast markers. What about the community of amateurs, taxonomists or professionals who don't have access to these tools? it's a question that has been asked many times but is still relevant.

Experimental design

very good

Validity of the findings

Very good, I appreciated the taxonomic conclusion, combined with the remark about the difficulty of integrating cryptic species, for example in a flora or in keys.

Additional comments

This paper can be accepted

Annotated reviews are not available for download in order to protect the identity of reviewers who chose to remain anonymous.

Reviewer 5 ·

Basic reporting

acceptable

Experimental design

acceptable

Validity of the findings

acceptable

Additional comments

Line 68 ‘Greasewort’ - this common name is superfluous here
Lines 91-93 ‘specimens, which were later corroborated by the molecular evidence using the ISSR method showing several cryptic species within A. pinguis (Buczkowska et al., 2016). Finally, Wawrzyniak et al. (2014) reported’ – this means the paper published in 2014 is successor of the paper published in2016 that is formally not correct.
Lines 64-84: I guess in this paragraph it should be indicated that no accessions of true A. maxima from Indonesian specimens were involved into any studies and the coincidence of the original and derived concepts is not obvious
Lines 279-281: the GenBank number for epitype should be provided here
Line 370: ‘A. maxima’ should be italicized.
Figure 2, right lower corner: ’outgroup’, not ‘outgroups’.
Finally I would be happy to see author’s opinion in the discussion section, whether observed genetic diversity in combination with the small (or none) morphological difference is the result of stasis.

Reviewer 6 ·

Basic reporting

General comments: I was really enjoy reading this article. It is an interesting and important study in Aneura and provides basic information about the taxonomy and species delimitation question of one of the most difficult group, Aneura pinguis.

Experimental design

I don't have comments on the experimental design. The design is ok for this objective.

Validity of the findings

Previous reviewers have given very good and enough comments. I don’t have more on this manuscript. Only thing I am not sure is that whether the outcomes of manuscript are enough to publish in PeerJ considering the record of A. maxima still cannot be removed.

Additional comments

My comments and suggestions for details were listed below:


1. Line 58 “taxa from five genera (Furuki, 1991; Preußing et al., 2010; Söderström et al., 2016).” You lost the reference of RABEAU et al., 2017 file:///C:/Users/xiang/Downloads/397-Article%20Text-2009-1-10-20170208.pdf. In this article the new genus of Afroriccardia was published. Also you should list the all five genera.
2. Line 89 “Buczkowska, Adamczak and Baczkiewicz (2006) reported..” . if the number of authors are over three, you should use “first author et al., “
3. Line 212 pleas add the package name for “wilcox.test”.
4. Line 265 Figure S2 I suggest move the PCA plot from supporting materials to the main text. It is better if you can show both morphological and genetic results in the main text. A PERMANOVA analysis is needed for the PCA so you can quantity the differences between Clade 1 and Clade 4.

---

## Round 0.3 · accepted · Accept

I appreciate taking all comments into account, they improve a lot the article. The only point you will have to rephrase is in the sentences from lines 360-363: "the named taxa are distinguishable for taxonomists without molecular data as well" . My suggestion is: "without the molecular evidence" will be better. You can change that during further editorial work.